# New Synthetic Partial Discharge Calibrator for Qualification of Partial Discharge Analyzers for Insulation Diagnosis of HVDC and HVAC Grids

**DOI:** 10.3390/s23135955

**Published:** 2023-06-27

**Authors:** Abderrahim Khamlichi, Fernando Garnacho, Pascual Simón

**Affiliations:** 1Fundación para el Fomento de la Innovación Industrial, FFII-LCOE, Eric Kandel Street 1, 28906 Madrid, Spain; fernando.garnacho@ffii.es (F.G.); psimon@ffii.es (P.S.); 2Departamento de Ingeniería Eléctrica, Electrónica, Automática y Física Aplicada, ETSIDI, Universidad Politécnica de Madrid, 28006 Madrid, Spain

**Keywords:** PD calibrator, HFCT PD sensors, IEC 60270, TS IEC 62478, offline PD measurements, online PD measurements, continuous PD monitoring, insulation defects in HV grids

## Abstract

A synthetic partial discharge (PD) calibrator has been developed to qualify PD analyzers used for insulation diagnosis of HVAC and HVDC grids including cable systems, AIS, GIS, GIL, power transformers, and HVDC converters. PD analyzers that use high-frequency current transformers (HFCT) can be qualified by means of the metrological and diagnosis tests arranged in this calibrator. This synthetic PD calibrator can reproduce PD pulse trains of the same sequence as actual representative defects (cavity, surface, floating potential, corona, SF_6_ protrusion, SF_6_ jumping particles, bubbles in oil, etc.) acquired in HV equipment in service or by means of measurements made in HV laboratory test cells. The diagnostic capabilities and PD measurement errors of the PD analyzers using HFCT sensors can be determined. A new time parameter, “PD Time”, associated with any arbitrary PD current pulse *i(t)* is introduced for calibration purposes. It is defined as the equivalent width of a rectangular PD pulse with the same charge value and amplitude as the actual PD current pulse. The synthetic PD calibrator consists of a pulse generator that operates on a current loop matched to 50 Ω impedance to avoid unwanted reflections. The injected current is measured by a reference measurement system built into the PD calibrator that uses two HFCT sensors to ensure that the current signal is the same at the input and output of the calibration cage where the HFCT of the PD analyzer is being calibrated. Signal reconstruction of the HFCT output signal to achieve the input signal is achieved by applying state variable theory using the transfer impedance of the HFCT sensor in the frequency domain.

## 1. Introduction

The development of the transmission and distribution grids together with the constraints of building new HV overhead lines due to their environmental impact has increased the use of underground insulated HV cable systems. To improve the reliability of HV cable systems, different parameters must be monitored: the cable temperature is measured to identify possible hot spots and to enlarge the ampacity of the line, vibrations are monitored to detect possible mechanical stresses, and acoustic technics are used to detect possible short circuit location in HV lines.

PD monitoring using HFCT sensors has proven to be efficient to prevent insulation defects in HVAC cable systems [1,2]. Insulation defects in cable systems, air insulation substations (AIS), gas insulated substations (GIS) or lines (GIL), power transformers, HVDC converters, etc., can cause catastrophic consequences, such as blackouts with important incidences of lost profits, dangerous explosions, destructive fires, which can be avoided thanks to the early detection of PD current pulses that occur in incipient defects (see Table 1).

For today’s insulated transmission cables with lengths of hundreds of km, only HVDC cable systems are used [3,4]. One disadvantage of these cable systems, unlike HVAC cables, is that they do not have link boxes to place HFCT-type sensors every 500 m or 700 m, while in HVDC cables, a longer distance of around 10 km is required, as shown in Figure 1. Consequently, the sensitivity of HFCT sensors used for HVDC must be higher than that required for HVAC cable systems.

PD monitoring of HVDC cable systems is currently challenging [5,6,7]. The conventional identification of the type of defect involved in AC cable systems through phase-resolved PD pattern recognition, PRPD, Refs. [8,9] is not applicable in HVDC cable systems. Furthermore, when the voltage remains constant in an HVDC grid, the rate of PD pulses due to a defect, such as a cavity in the insulation, is negligible since they appear mainly when the voltage changes [7]. Even for commissioning and maintenance tests of HVDC cable systems, power frequency voltages are applied instead of direct voltage [10,11,12,13] because PD measurements are easier in HVAC than in HVDC.

The traveling charge through an HV transmission cable system remains nearly constant along the cable; for this reason, the charge magnitude measured by an HFCT sensor placed in the cable sheath is the key quantity for insulation diagnosis. The actual charge of PD traveling pulses can be determined by signal processing using HFCT sensors with appropriate bandwidths.

The main steps that a PD analyser for continuous monitoring of HVDC cable systems must perform for a suitable insulation diagnosis are presented in Figure 2. These requirements have been considered to design the tests to be implemented in the synthetic PD calibrator.

Measurements in real time (1) are needed for continuous PD monitoring in HVDC systems because PD pulses appear mainly during voltage changes, such as voltage polarity reversals and temporary or transient over-voltages as switching and lightning surges.

Noise signals coming from TV broadcasters, mobile phone stations, etc., must be mitigated by filtering (2) [14,15]. Only pulse signals with a pulse width within a specific time duration and with the same polarity as the applied voltage should be considered as possible PD pulses generated in the cable system or its accessories.

After noise suppression (2), the PD location of internal PD sources along each monitored cable section should be determined by means of two successive HFCT sensors (3) [16,17,18,19]. The classic analysis of traveling waves is an effective method to determine where a PD source is placed along a cable system. Synchronized PD measuring systems placed every two consecutive link boxes where cable sheaths are accessible are used to detect eventual PD pulses. Pulse pairs acquired by two synchronized HFCT sensors, S_1_ and S_2_ (see Figure 1), whose arrival time delay, Δ*t*, is lower than the time required for travelling the distance between both HFCT sensors, *L*, means that the PD source is located between both sensors. Using Δt and the propagation velocity, v, along the cable system, the distance, x, to the left sensor can be determined by Formula (1)
(1)$$x=\frac{L-\Delta t\cdot v}{2}$$

If Δt=L/v, it means that the pulse has travelled the length of the cable section that separates both consecutive HFCT sensors, *L*, then the defect would be located at the vicinity of a cable accessory, or outside the monitored cable section. The pulse width will indicate if the pulse is close to or far from the sensors. A pulse width of less than a few tens of nanoseconds means that the defect is close to the sensor: the wider the pulse is, the farther the defect is from the sensor.

The detection of insulation defects close to the HVDC substation is more difficult because it requires power clustering tools (4) to discriminate different PD sources and pulsating noise. Pulsating noise due to power electronics such as thyristors or IGBTs of rectifiers and converters, especially disturbing close to cable ends, must be separated from PD pulse signals using clustering tools (4). Clustering tools by frequency analysis of each pulse or waveform time parameters [20,21] are useful to separate PD sources from pulsating noise signals. PD recognition tools (5) [22,23,24,25,26,27,28] based on PD histograms, accumulated charge vs. time or other PD patterns are used to discriminate different insulation defects: corona, surface PD, floating, etc.

This paper presents a synthetic PD calibrator to qualify PD analysers using HFCT sensors through two different types of tests: metrological and diagnosis tests. Metrological tests allow the evaluation of errors due to noise influence, linearity errors, measuring errors due to different pulse widths, and a lower resolution time between PD pulses. Diagnosis tests allow the analysis of different PD capabilities: location, clustering, and defect recognition.

## 2. PD Pulse Waveforms Generated by Insulation Defects

The actual PD pulse generated in an insulation defect, for example, within a cavity, depends on the physics of the discharge in the gas cavity: the molecular composition, pressure and temperature, discharge gap, insulation material, etc., resulting in current waveforms of a few nanoseconds width [29]. A cable system works as a low-pass band filter distorting the current PD pulse *i(t)* that travels along the cable, filtering out its high-frequency content and therefore increasing its width. The expected distortion can be estimated by theoretical analysis using the spectrum of the HVDC cable system transfer impedance, *Z_c_(f)*; see [30,31].

An inverse double exponential (IDE) function (1) with known *α* and *β* parameters and slope zero in its origin can be used to emulate actual PD pulses with arbitrary time parameters *T_1_/T_2_*. In Figure 3, some different PD pulse waveforms are shown with their associated *α* and *β* parameters: (a) a reference PD pulse showing the definition of the time parameters *T_1_/T_2_*, (b) an asymmetric PD pulse with short front time, (c) an asymmetric PD pulse with a long front time, and (d) a symmetric PD pulse.
(2)$$i\left(t\right)=i_{peak}\cdot k\cdot \frac{1}{e^{\alpha \cdot t}+e^{-\beta \cdot t}}$$
where:
-$k=\frac{\beta +\alpha }{\beta }\cdot {\left(\frac{\beta }{\alpha }\right)}^{\frac{\alpha }{\beta +\alpha }}$-*i_peak_*: peak value of the PD pulse *i(t)*

**Figure 3 sensors-23-05955-f003:**
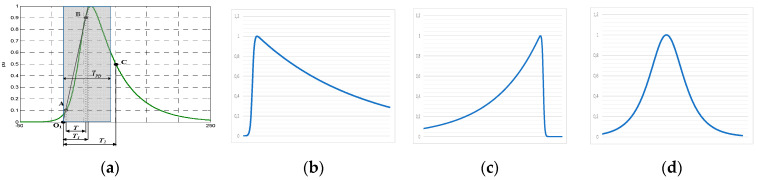
Inverse DE waveforms used to emulate any PD pulse: (**a**) reference PD pulse with 1/α = 44 ns; 1/β = 9.9 ns, *T*_1_/*T*_2_ = 37.8/82.0 ns, *T_PD_* = 75 ns; (**b**) short front time asymmetric PD pulse with 1/α = 35.7 ns; 1/β = 0.3 ns, *T*_1_/*T*_2_ = 1.7/27.4 ns, *T_PD_* = 37.5 ns; (**c**) long front time asymmetric PD pulse with 1/α = 0.5 ns; 1/β = 34.8 ns, *T*_1_/*T*_2_ = 89.3/86.7 ns, *T_PD_* = 37.5 ns; (**d**) symmetric PD pulse with 1/α = 11.9 ns; 1/β = 11.9 ns, *T*_1_/*T*_2_ = 36.1/53.8 ns, *T_PD_* = 37.5 ns.

The charge, q, of any PD current pulse can be expressed by means of the “PD time” parameter, *T_PD_*, and its peak value, *i_peak_*.
(3)$$q=\int_{0}^{\infty }i\left(t\right)\cdot dt=i_{peak}\cdot \int_{0}^{\infty }\frac{i\left(t\right)}{i_{peak}}\cdot dt=i_{peak}\cdot \int_{0}^{\infty }i_{pu}\left(t\right)\cdot dt=i_{peak}\cdot T_{PD}$$

The PD time parameter, *T_PD_*, is defined as the width of the equivalent rectangular pulse that has the same charge, q, and current peak, $i_{peak}$, values as the original current PD pulse. The *T_PD_* of a PD pulse following an IDE function can be easily determined by (4), using the trigonometric function cosecant (csc).
(4)$$T_{PD}=\frac{\pi }{\beta }\cdot {\left(\frac{\beta }{\alpha }\right)}^{\frac{\alpha }{\beta +\alpha }}\cdot \csc\left[\frac{\pi \cdot \alpha }{\alpha +\beta }\right]$$

Consequently, the *T_PD_* value of a PD pulse represents the charge magnitude of a current pulse, *i(t)*, if its amplitude, $i_{peak}$, were the unit. For example, a pulse with a PD time *T_PD_* of 75 ns represents a pulse with a charge value of 75 pC if its peak value is 1 mA.

The cut-off frequency (−3 dB), *f_c_*, of PD pulses following an IDE depends on both *α* and *β* parameters as shown by Equation (5). The PD time, *T_PD_*, is a parameter that is very closely related to the cut-off frequency of the pulse *f_c_*. Table 2 shows that despite very different values of *α* and *β* parameters, the cut-off frequency is well represented by *T_PD_*.
(5)$$f_{c}=\frac{\alpha +\beta }{2\cdot \pi^{2}}\cdot a\sinh\left[\sin(\frac{\pi \cdot \alpha }{\alpha +\beta })\right]$$

## 3. Study of PD Pulse Widths in HVDC Cable Systems to Define HFCT Sensor Characteristics

The scale factor is defined by IEC 60270 [32] as the factor to be multiplied by the instrument reading to obtain the charge quantity. For example, for a measuring system whose instrument reading is expressed in peak voltage, mV, the scale factor is expressed in pC/mV.

If the transfer impedance of the HFCT sensor, *Z_s_*, remains almost constant for all pulse frequency spectra with a 50 W load at its output, the scale factor can be calculated by:(6)$$k\left(\frac{pC}{\mathrm{mV}}\right)=\frac{q}{u_{peak}}\approx \frac{i_{peak}\cdot T_{PD}}{Z_{s}\cdot i_{peak}}=\frac{T_{PD}\left(\mathrm{ns}\right)}{Z_{s}\left(\frac{\mathrm{mV}}{\mathrm{mA}}\right)}=\frac{1}{s}$$

The scale factor, *k* depends on the transfer impedance, *Z_s_*, of the HFCT and on the *T_PD_* of the PD pulse to be measured. For very short pulses with a broadband frequency spectrum, the transfer impedance, *Z_s_*, may not remain constant, and the scale factor will not follow Formula (6). The sensitivity, s, of the measuring system is the inverse of its scale factor, *k*; the higher the *mV* output of the sensor for the same charge value of *pC*, the higher the sensitivity is.

Figure 4a shows the attenuation and PD distortion of a PD pulse initially generated with a *T_PD_* = 5 ns when traveling along an HVDC cable system of U_0_ = 320 kV, as shown in Figure 4b, which demonstrates the growth of *T_PD_* versus the traveling distance. It can be observed that the pulse *T_PD_* increases up to 375 ns after traveling about 11 km and its peak value decreases in the same proportion, keeping the electrical charge of the traveling current pulse almost constant.

This means that the same traveling charge due to an insulation defect in a cable system can be measured on the cable sheath at any distance along the cable but decreases its pulse amplitude as the pulse travels (see Figure 4a). The worst case for detecting the PD pulse is at the longest distance to the defect, where the pulse voltage peak, $u_{p,min}$, can still be measured with enough accuracy.
(7)$$q_{min}=u_{p,min}\cdot \frac{T_{PD,\;\;L_{max}}}{Z_{s}}$$
where:
-$u_{p,min}$: minimum pulse voltage peak that can be detected.-*q_min_*: minimum charge value that the PD analyser can measure.-*Z_s_*: transfer impedance of the HFCT sensor.-$T_{PD,\;\;L_{max}}$: PD time of the PD pulse to be measured at the longest distance, $L_{max}$.


If a PD sensitivity of 20 pC were required for a continuous PD monitoring system, a 10 pC pulse would travel to each side of the cable system. These pulses should be measured with the HFCT sensors placed in the link boxes. Considering 11 km between two consecutive HFCT sensors, and that the defect could be near one sensor, the pulse would arrive to the other HFCT sensor with a *T_PD_* = 375 ns and with a peak value of 0.027 mA (10 pC/375 ns= 0.027 mA). Assuming a peak voltage sensitivity of an average PD monitoring analyser of at least 0.4 mV, the required transfer impedance of the HFCT sensor, *Z_s_*, to be used with the PD analyser would be:(8)$$Z_{s}=U_{p.min}\cdot \frac{T_{PD}}{q_{min}}=0.4\;\mathrm{mV}\cdot \frac{375\;\mathrm{ns}}{10\;\mathrm{mA}\cdot \mathrm{ns}}=15\frac{\mathrm{mV}}{\mathrm{mA}}=15\;\Omega$$

Commercial HFCT sensors used for HVAC cable systems typically have *Z_s_* values in the range of 4 mV/mA to 12 mV/mA. Consequently, they are not sensitive enough to be used in HVCD cable systems with distances between HFCT sensors up to 11 km.

## 4. PD Analysers for Insulation Diagnosis of HV Cable Systems

### 4.1. PD Analyser Working according to IEC 60270

A PD measuring system according to IEC 60270 consists of a measuring impedance, *Z_m_*, as a coupling device, a coaxial cable as a transmission system, and a PD measuring instrument (see Figure 5) working in a frequency range not exceeding 1 MHz. The method of IEC 60270 to determine apparent charge values is by a quasi-integration approach. It consists of transforming the current pulse *i(t)* into a voltage pulse, *u(t)* at the terminals of the measuring impedance, Z_m_, whose peak value is proportional to the charge magnitude, q, of the PD current pulse *i(t)*, according to Formula (6); see [33]. The measuring impedance works as a passband filter for PD current pulses (see Figure 6).
(9)$$u_{peak}=2\cdot \left(f_{2}-f_{1}\right)\cdot Z_{m}\cdot q$$
where:
-$f_{2}$: upper limit frequency of the measuring transfer impedance, for which its value does not differ by more than 6 dB from the rated transfer impedance.-$f_{1}$: lower limit frequency of the measuring transfer impedance, for which its value does not differ by more than 6 dB from the rated transfer impedance.-$Z_{m}$: the impedance gain of transfer impedance Z(s) in the flat zone between $f_{1}$ and $f_{2}$.

**Figure 5 sensors-23-05955-f005:**
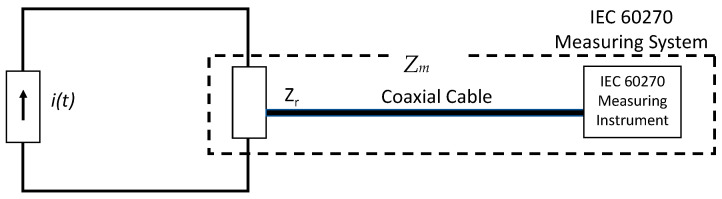
Measuring system according to IEC 60270.

**Figure 6 sensors-23-05955-f006:**
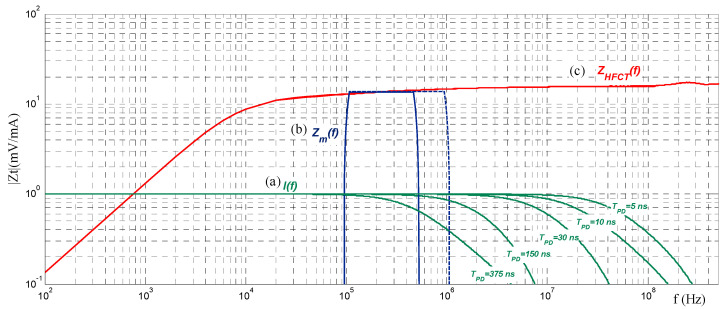
Different approaches to measure PD pulses: (a) *I(f)* = normalized spectrum of different PD pulses from 5 ns to 375 ns; (b) *Z_m_(f)*: measuring system according to IEC 60270 (*f*_1_ = 100 kHz, *f*_2_ = 0.5 MHz) and the *f*_2_ extension up to 1 MHz (future IEC 60270); (c) measuring system using HFCT sensor with a flat response from 100 kHz up to 100 MHz.

To ensure that the PD current pulse spectrum remains flat in the frequency range $\Delta f=f_{2}-f_{1}$ at which PD pulse is measured, the current IEC 60270 establishes the upper cut-off frequency $f_{2}$ of the passband filter of not more than 0.5 MHz (see Figure 6). Therefore, according to IEC 60270, the frequency spectrum of the measuring impedance, *Z_m_*, should be shorter than the frequency spectrum of the PD current pulse *I(f)*. For very fast transient pulses with pulse widths of a few tenths of a nanosecond, this requirement is easily met, but for PD pulses with long *T_PD_* values, the quasi-integration error could potentially not be negligible because the PD pulse spectrum is not flat in the measuring frequency range Δf.

### 4.2. PD Analyzers according to TS IEC 62478 Using HFCT Sensors

The technical specification TS IEC 62478 [34] deals with measurements of partial discharges by electromagnetic and acoustic methods, including as one of these methods using HFCT sensors as coupling devices, together with a coaxial cable as the transmission system and a PD measuring instrument as shown in Figure 7.

A measuring system using an HFCT acquires most of the pulse frequency content of the measured PD pulse (see Figure 6), in such a way that the original pulse can be reconstructed (see Section 5). For this reason, unlike an IEC 60270 measuring system, the upper cut-off frequency f_2_ of the HFCT transfer impedance, $Z_{s}$, must be higher than the upper-frequency limit fc of the pulse spectrum. The PD sensitivity of an HFCT sensor is related to its transfer impedance value, $Z_{s}$. The higher the value of the transfer impedance, the higher the sensor sensitivity.

Even if IEC TS 62478 considers a high-frequency range from 3 MHz to 30 MHz, the reference measuring system of the synthetic calibrator enlarges this range from 0.07 MHz to 200 MHz to minimise the distortion of the PD current pulses to be measured; it does not matter if the PD source is closed to the HFCT or several km away the sensor. For instance, the expected cut-off frequencies of pulses with a *T_PD_* from 10 ns to 375 ns are between 0.5 and 24.6 MHz (see Figure 6).

The transfer impedance spectra of several commercial HFCT sensors (A, B, C, D) are shown in Figure 8, with $Z_{s}$ values in the range from 4 mV/mA to 12 mV/mA. These transfer functions are suitable for HVAC cable system continuous PD monitoring but not for HVDC cable systems.

To overcome this drawback, an improved HFCT sensor with a transfer impedance of 15 mV/mA (sensor Type E of Figure 8) was developed using a special nanocrystalline ferrite core with a 3 cm external diameter and special flat-shaped copper conductor for the winding. According to IEC 60270, the transfer impedance bandwidth limits of −6 dB, *f_1_* and *f_2_*, are 20 kHz and 500 MHz, respectively. Nevertheless, the transfer impedance is flat at 15 mV/mA ± 5% only in the frequency range between *f_A_* = 0.5 MHz and *f_B_* = 200 MHz. Table 3 summarize the transfer impedance characteristics of the commercial HFCT sensors A, B, C, D and E.

## 5. Synthetic PD Calibrator

The PD synthetic calibrator consists of an arbitrary waveform generator (AWG) with a reference measuring system composed of a specially designed HFCT sensor connected to a digital recorder. This calibration setup generates PD pulses, measures its current, and determines its electrical charge by signal processing (see Section 5.5).

### 5.1. Electrical Circuit

The developed synthetic PD calibrator follows the electrical circuit of Figure 9. It consists of an arbitrary waveform generator, AWG, (1), with a 400 MHz bandwidth, 1.25 Giga-Samples/s, and a 50 Ω internal resistance (2) that reproduces PD pulses in a pre-defined sequence according to a PD event train chosen from a reference database (3) to generate a calibration PD pulse train or to emulate a real PD pulse train representative of an insulation defect (e.g., a cavity, corona, surface or floating). Each PD event train that lasts several seconds up to several minutes is an array of charge values together with their starting times ($q_{i},t_{i})$.

A pulse train is generated using the chosen PD event train, giving the same waveform to each charge event, ($q_{i},t_{i}),$ by means of the analytical functions described in Section 2 and saved in the reference pulse waveform database (4). The generated PD pulses are injected in a current loop matched at 50 Ohm with a terminal load resistance of 50 Ohm, $R_{load}$, (5).

Each generated PD current pulse is acquired through two improved HFCT sensors, (6) and (7), with a high transfer impedance value of 15 mV/mA each, placed before and after the open testing cell (8) where the HFCT sensor of the PD analyzer under characterization must be placed. The two sensors are used to reduce uncertainty and to check that the HFCT under test is properly matched, preventing signal reflections. The output signals of HFCT sensors (6) and (7) are measured by a digital recorder with a 200 MHz bandwidth (1 Giga-Sample/s with an 8-bit resolution or 0.5 GSamples/s with 12 bits resolution) (9). A PC (12) is used to upload the reference PD event trains (3) with a determined reference pulse waveform (4). The PC is also used for pulse reconstruction (10) and integration (11) by means of the signal processing software described in 5.4. The resulting charge (11) is used as feedback to regulate the voltage amplitude of the AWG to achieve the PD pulse charge previously set.

This PD calibrator can be used for “metrological tests” and “diagnostic tests” of PD analyzers. The metrological setting is used to evaluate the following characteristics of the PD analyzer under test: (1) errors caused by noise, (2) linearity errors, (3) errors due to different pulse widths, and (4) the resolution time. The diagnostic setting evaluates PD clustering, PD recognition, and PD location capabilities. Metrological tests are traceable to national standards, while diagnostic tests are referred to a database of actual insulation defects.

### 5.2. Reference PD Pulses and Reference PD Pulse Trains

When using the setting of “metrological tests” the synthetic PD calibrator generates PD pulses following the IDE function according to (2). Considering Table 2 and Figure 4b, the cut-off frequency range of the actual PD pulses can change by 7 MHz when the sensor is less than 1 km from the fault source to around 0.5 MHz when the sensor is 11 km away from the PD source; a cut-off frequency of 3.3 MHz would be between both extreme cut-off frequencies, and therefore, an IDE pulse with *T_PD_* of 75 ns and *T*_1_/*T*_2_ = 31.2/76 ns (see Figure 10) is used as the reference PD current pulse for most of the metrological tests.

When using the setting of “Qualification of Diagnostic Capabilities” the synthetic calibrator generates damped oscillating PD pulses, simulating real PD pulses that oscillate due to circuit inductances.
(10)$$i\left(t\right)=i_{peak}\cdot \sin\left(2\cdot \pi \cdot f\cdot t+\varphi \right)\cdot k\cdot \frac{1}{e^{\alpha \cdot t}+e^{-\beta \cdot t}}$$

This sinusoidal waveform with the oscillation frequency, *f*, and a phase shift, *φ*, is damped by an IDE function. The time and frequency parameters of calibration and diagnosis pulses are shown in Table 4 and Table 5, respectively.

A train of PD pulses representative of an insulating defect is emulated by means of one of the four PD pulses referred to in Table 5 (Figure 11b) generated following a PD event train in a defined sequence of charge values and starting times ($q_{i},t_{i})$.

### 5.3. Practical Implementation

A general overview of the developed synthetic calibrator is shown in Figure 12. The AWG (1) and the PC (2) are installed inside the same metallic envelope. The open test cell (8) is used to place the HFCT sensor under characterization (15). The two improved HFCT sensors included in the synthetic PD calibrator are (6) and (7). To avoid signal reflections between these elements connected in series by coaxial cables, the geometry of the open test cell (8) and the internal configuration of the HFCT sensors are dimensionally designed to approach a characteristic impedance of 50 Ω. The current loop is closed with a 50 Ω coaxial load resistance (5) to achieve impedance matching. A digital recorder (9) with a 250 MHz bandwidth is placed below the open test cell (8) to acquire output voltages from the reference HFCT sensors (6) and (7). A computer keyboard (13) with a built-in screen (14) is available as a user interface to control and manage the qualification tests.

The AWG generator can use an internal memory of 4 GB or an external memory of 240 GB. When the external memory is used, the PD pulse generation is carried out in streaming mode with a maximum transferring speed of 100 MS/s. Bearing in mind that, due to the resolution, 2 bytes are required to manage each sample, up to 20 min of PD pulse generation can be played (240 GB/100 MS/s/2 B/S). A sampling interval of 10 ns is appropriated for the diagnosis qualification of PD measuring systems with HF range up to 30 MHz.

However, for some metrological tests, the maximum data transfer capacity that the calibrator can generate (up to 1.2 GS/s) is needed, which is only possible using the internal memory of 4 GB. For example, to generate PD pulses with a very short *T_PD_* of 8 ns, about 30 samples spaced at 0.8 ns are used. At the maximum data transfer capacity speed, a length record of 1.6 s can be played. For other metrological tests, a transfer rate of 1 GS/s is used to generate PD pulse trains of 2 s length.

For AC (50 Hz), each individual PD pulse train of 2 s consists of 100 periods of 20 ms, to generate a resolved phase PD pattern well representing a defect type (cavity, floating potential, etc.) to be identified by any expert technician. At least 500 PD pulses are needed to represent the PRPD pattern of a real defect, but at the same time, for memory limitation, the maximum number of DP pulses for each PD train representing an insulation defect is limited to 4000. Considering that each PD train must be generated in 2 s, the PD repetition rate of any insulation defect must be in the range of 5 pulses/period to 40 pulses/period. This figure can be regulated depending on the defect type or insulation aging degree to be simulated.

To simultaneously reproduce more than one defect, e.g., for the PD clustering test in AC, up to four PD trains of 2 s can be overlapped. Each AC period of 20 ms has 10,000 intervals of 2 μs where a PD pulse can be placed. If two pulses from different PD trains happen to coincide in the same 2 μs interval, one of the pulses must be shifted to the next 2 μs interval.

### 5.4. Reference PD Measuring System

The reference PD measuring system is based on two improved HFCT sensors, a digital recorder, and signal processing software to determine the charge value of the resulting signal after its reconstruction.

The maximum open circuit peak voltage of the AWG is 4 V, which, through a current loop of 100 Ω (see Figure 9), corresponds to a maximum peak current, *i_peak_*, of 40 mA. According to Formula (3), the maximum transferred charge will depend on the *T_PD_* of the generated pulse, for example, if *T_PD_* = 75 ns, the maximum transferred charge will be 75 ns × 40 mA = 3.0 nC. Other maximum charge values can be generated depending on the *T_PD_* (see Table 6). The lowest charge values are limited by the measurement sensitivity of the digital recorder, which is the minimum peak value that the digital recorder can measure with specified uncertainty (see Section 6). Assuming a minimum peak value of 0.4 mV for the recorder and an HFCT transfer impedance, $Z_{s}=15\;\mathrm{mV}/\mathrm{mA}$, the minimum charge value can be detected depending on the *T_PD_* values (see Table 6), for instance, if *T_PD_* = 75 ns, the minimum charge value would be 2 pC [0.4 mV/(15 mV/mA) × 75 ns =2 pC].

### 5.5. Reconstruction of the Original PD Pulse and Signal Integration

The reconstruction of the original PD pulse measured by the improved HFCT sensors is required to determine the electrical charge of the current pulses at the HFCT input. The transfer function in the frequency domain of the HFCT sensor must be previously determined by a characterization test (see Figure 13). The HFCT transfer function can be fitted [35] by expression (11) as products of quotients of poles and zeros or by expression (12) as the sum of poles with their residuals. The transfer function of the reference HFCT sensors was fitted by Formula (12) with eight poles and nine residues (Figure 13). It can be observed that the fitted curves are overlapping the measured ones.
(11)$$Z_{t}\left(s\right)=\prod_{i=1}^{n}\frac{s-z_{i}}{s-p_{i}}$$
(12)$$Z_{t}\left(s\right)=\sum_{i=1}^{n}\frac{r_{i}}{s-p_{i}}+r_{0}$$

According to state theory applied to continuous-time systems [36], the current PD pulse at the output of the sensor terminals, *u(t)*, can be determined in the time domain by means of state variable *x(t)*.
(13)$$\begin{array}{c} \dot{x}\left(t\right)=A\cdot x\left(t\right)+B\cdot i\left(t\right)\\ u\left(t\right)=C\cdot x\left(t\right)+D\cdot i\left(t\right) \end{array}$$
where
(14)$$A=\left[\begin{array}{ccc} p_{1} & \cdots & 0\\ \vdots & \ddots & \vdots \\ 0 & \cdots & p_{n} \end{array}\right]$$
(15)$$B=\left[\begin{array}{c} 1\\ \ldots \\ 1 \end{array}\right]$$
(16)$$C=\left[\begin{array}{ccc} r_{1} & \ldots & r_{n} \end{array}\right]$$
(17)$$D=r_{0}$$

*n:* order of the model

*x(t)*: state-space vector of n × 1 dimension

*i(t)*: input current at the HFCT

*u(t)*: output signal of the HFCT

*A*: state matrix of n × n dimension

*B*: input vector of n × 1 dimension

*C*: output vector of 1 × n dimension

*D*: constant term = *r_o_*

The continuous-time system represented by equation system (13) can be transformed to its equivalent discrete system [36] by the integral approximation method (18). Considering a constant interval sampling $h_{s}$, this equivalent discrete system becomes:(18)$$\begin{array}{c} x_{k+1}=F\cdot x_{k}+G\cdot i_{k}\\ u_{k+1}=C\cdot x_{k+1}+D\cdot i_{k+1} \end{array}$$
where
(19)$$x_{0}=0$$
$$F=\exp\left(A\cdot h_{s}\right)$$
(20)$$G=\left(F-I\right)\cdot A^{-1}\cdot B$$

*I*: n × n identity matrix

Equation (18) can be transformed in the following expression:(21)$$u_{k+1}=C\cdot F\cdot x_{k}+C\cdot G\cdot i_{k}+D\cdot i_{k+1}$$

Adding and subtracting the term $D\cdot i_{k}$, the above equation becomes:(22)$$u_{k+1}=C\cdot F\cdot x_{k}+\left[C\cdot G+D\right]\cdot i_{k}+D\cdot (i_{k+1}-i_{k})$$

Assuming $D\cdot (i_{k+1}-i_{k})$ is small enough that it can be considered $\left[C\cdot G+D\right]\cdot i_{k}\gg D\cdot (i_{k+1}-i_{k}),$ then
(23)$$C\cdot G\cdot i_{k}+D\cdot i_{k+1}\approx \left[C\cdot G+D\right]\cdot i_{k}$$

And Equation (21) is transformed into the following:(24)$$u_{k+1}\;C\cdot F\cdot x_{k}+\left[C\cdot G+D\right]\cdot i_{k}$$

Consequently, any sample of the input current, $i_{k}$, in a generic *k*th sampling interval is related to the same *k*th sample of the voltage at the HFCT output sensor, by the following expression:(25)$$i_{k}=\frac{\left[u_{k+1}-\left(C\cdot F\right)\cdot x_{k}\right]}{\left(C\cdot G\right)+D}$$
where
(26)$$x_{k}=F\cdot x_{k-1}+G\cdot i_{k-1}$$

This reconstruction approach has been applied to three different signals with *T_PD_*= 7 ns, 37 ns, and 300 ns, generated by means of an arbitrary wave generator, the output of which was simultaneously measured by means of a reference digital recorder, *u_ref_ (t)*, and by the improved HFCT sensor used in the synthetic PD calibrator, *u(t)*. This HFCT output signal is processed according to Formula (25) to reconstruct the input current signal, *i(t)*. To compare the three waveforms, *u_ref_ (t)*, *u(t)*, and *i(t)*, all of them are expressed per unit of the peak value: *u_ref,pu_(t)*, *u_pu_(t)*, and *i_pu_(t)*. The voltage measured by the digital recorded, *u_ref,pu_(t)*, can be considered as the reference current, *i_ref,pu_ (t)*, because the internal impedance of the generator and the input impedance of the recorded have the same value and flat frequency response.

Figure 14 shows that the HFCT output, *u_pu_(t)*, fits quite well *i_ref,pu_ (t)* because the improved HFCT has a transfer impedance with a very flat frequency response. It is also observed that waveform *i_pu_(t)* fits perfectly, *i_ref,pu_ (t)*, proving the robustness of the signal reconstruction method.

The current integration of the reconstructed PD pulse can be calculated in the time domain or in the frequency domain. The frequency domain integration determines the frequency spectrum *I(s)* of the reconstructed pulse *i(t)* and uses, as the integral of the current signal, the limit value of *I(s)* for *s* = 0. The integration in the time domain is calculated by means of the trapezoidal rule.

## 6. Characterization of the Synthetic PD Calibrator

### 6.1. Error of the Generated Charge for Different PD Pulse Widths

To evaluate the performance of the synthetic PD calibrator, different current waveforms of the same charge value have been generated.

According to Formula (2), three parameters define any current signal *i(t)* fitted by an IDE waveform (*α*, *β*, and $i_{peak}$). Using these inputs, the charge value can be calculated by Formula (3), as the product of *T_PD_* and $i_{peak}$; therefore, the influence of these two parameters in the generated charge is evaluated.

The error provoked by the $i_{peak}$ parameter depends on the self-heating of the synthetic generator; however, in practice, this effect is minimized by means of the feedback loop with the HFCT sensor output. This feedback signal allows the AWG output voltage to be changed to achieve the target *i_peak_* value with less than 1% error.

To evaluate the error caused by the *T_PD_* parameter, the area of the reconstructed *i_pu_(t)* signal is calculated by signal integration, and this result is compared with the chosen *T_PD_* parameter in the generator to determine the error. This error is determined for *T_PD_* values from 8 ns to 375 ns (see Table 7). The assigned uncertainty of the PD synthetic PD calibrator for charge measurements of ±2% or 1 pC (whichever is greater) is compatible with the determined charge errors.

### 6.2. Linearity Test

For the linearity characterization of the measuring system integrated into the synthetic PD calibrator, its generated PD pulse trains with equal PD pulse amplitudes and the same PD pulse waveforms. Each pulse train is made up of consecutive bursts of four pulses every 10 ms (n = 400 pulse/s) separated by 1 ms between them. The results of the measuring system integrated into the synthetic PD calibrator were compared with the measurement performed with an IEC 60270 measuring system. The IEC 60270 measuring system was set at 200 pC amplitude using the synthetic PD calibrator. PD pulse trains of the same *T_PD_* value but with different charge values were injected in the ranges of 2.5 nC, 1.25 nC, 500 pC, 100 pC, 50 pC, 10 pC, and 2 pC without anyº superimposed noise. The charge measurement errors with respect to the reference values given by IEC 60270 measuring system are shown in Table 8 expressed as percentage of the injected charge and in the pC. The linearity figures have been calculated as half the difference between the maximum and minimum charge measurement errors.

## 7. Conclusions

The developed PD synthetic PD calibrator is an arbitrary wave generator that plays current PD pulse trains. Every current pulse of a PD train is made by means of an inverse double exponential function of known time parameters 1/α and 1/β. The generated PD current pulse is acquired by two identical sensitive HFCT sensors, whose transfer impedance spectrum was previously characterized in terms of the amplitude and phase shift for a frequency range between 100 Hz and 500 MHz.

The voltage signal at the output of each HFCT sensor is measured by a 200 MHz bandwidth digital recorder with a sampling rate of 0.5 GS/s operating at 12 bits (1 GS/s operating at 8 bits). Next, the original current signal at the input of each HFCT sensor, *i(t)*, is reconstructed by means of a state variable model, using the transfer function of the HFCT sensor, and the voltage signal at the output of each HFCT sensor, *u(t)*. The charge quantity of each PD pulse, *q*, is finally determined by applying the final value property of the Laplace transform to the function *I(s)*, obtained from the reconstructed current signal *i(t)*.

This synthetic PD calibrator can reproduce current pulse trains of stable charge values from 2 pC to 15 nC with an uncertainty of less than ±2% or ±1 pC, whichever is greater, and with a time separation between pulses not less than 5 μs.

This calibrator will help to improve the functionality and metrological accuracy of PD measurement instruments by means of its adequate and simple characterization and calibration, providing better supervision of the insulation condition of high-voltage assets (cables, transformers, generators, or GIS systems) through the improvement of PD measuring instruments and analyzers.

## Figures and Tables

**Figure 1 sensors-23-05955-f001:**
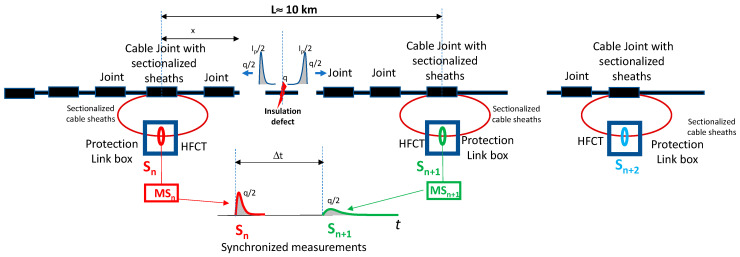
Distribution of link boxes along an HVDC cable system.

**Figure 2 sensors-23-05955-f002:**
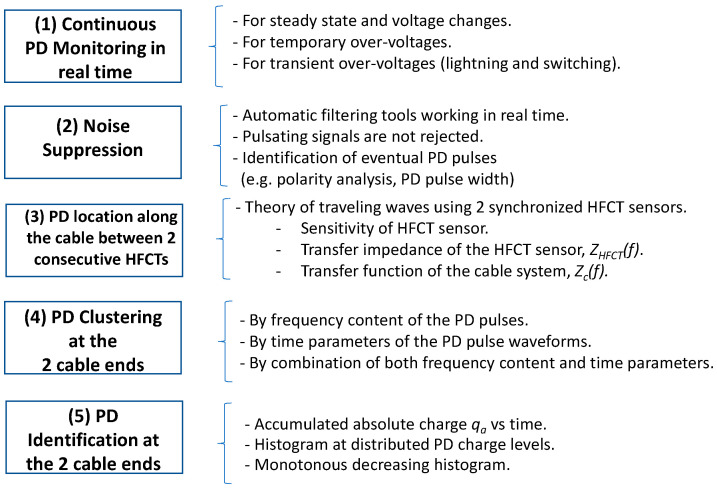
Steps performed by an HVDC monitoring system that uses HFCT sensors.

**Figure 4 sensors-23-05955-f004:**
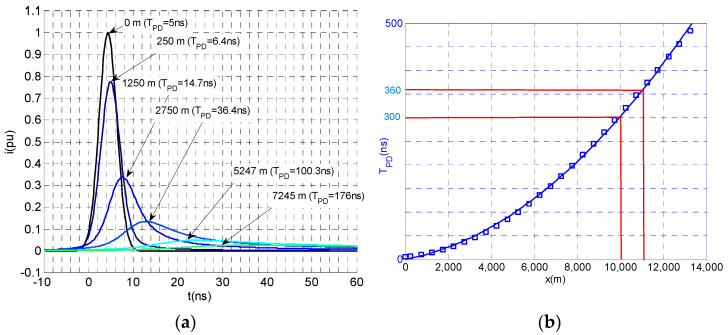
(**a**) PD pulse distortion traveling along an HVDC cable system. (**b**) Growth of the PD time, *T_PD_*, traveling along an HVDC cable system with an initial *T_PD_* = 5 ns.

**Figure 7 sensors-23-05955-f007:**
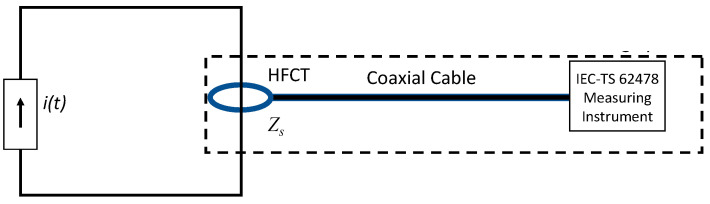
Measuring system using an HFCT as a coupling device (IEC-TS 62478).

**Figure 8 sensors-23-05955-f008:**
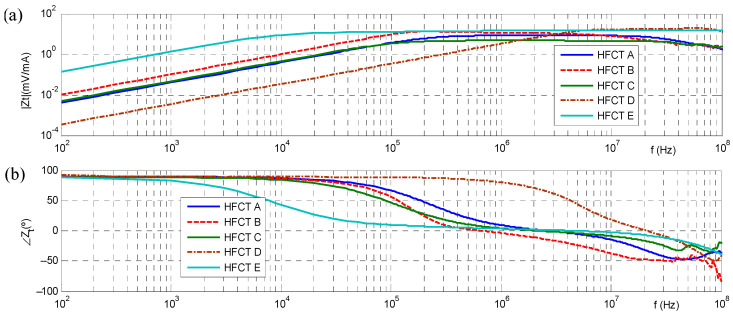
Transfer impedance of commercial HFCT sensors: (**a**) amplitude of the transfer impedance Z_t_ expressed in (mV/mA); (**b**) phase shift of the transfer impedance expressed in degrees (°).

**Figure 9 sensors-23-05955-f009:**
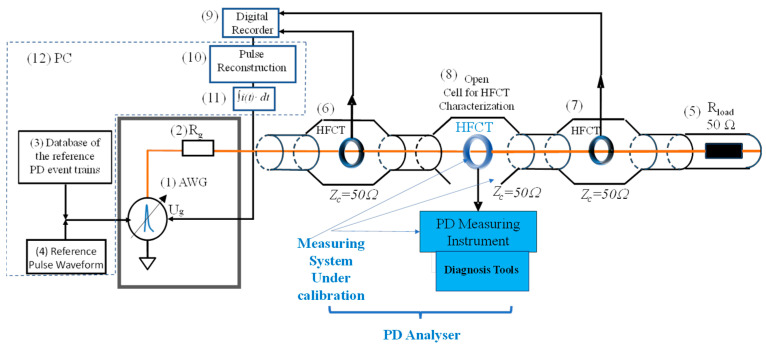
Components of the reference synthetic calibrator.

**Figure 10 sensors-23-05955-f010:**
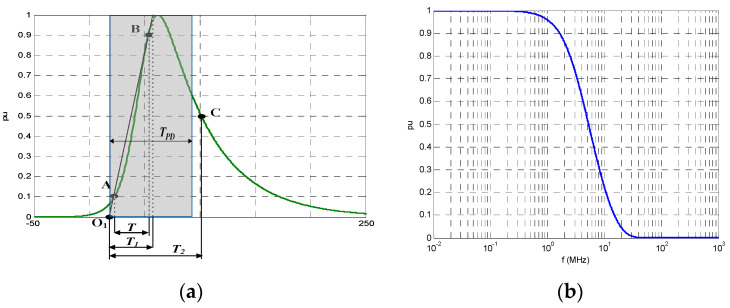
PD pulse waveform of *T_PD_* = 75 ns using the IDE function (1/α = 44 ns and 1/β = 9.9 ns). (**a**) Waveform in the time domain; (**b**) amplitude spectrum in the frequency domain.

**Figure 11 sensors-23-05955-f011:**
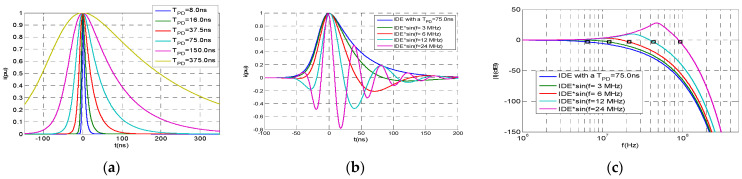
Reference PD pulses for PD analyzer qualification: (**a**) PD pulse waveforms used for metrological tests with different T_PD_; (**b**) damped oscillating PD pulse waveforms used for qualification of diagnostic capabilities; (**c**) frequency spectrum for damped oscillating PD pulse waveforms, with marks for the −3 dB cut-off frequency.

**Figure 12 sensors-23-05955-f012:**
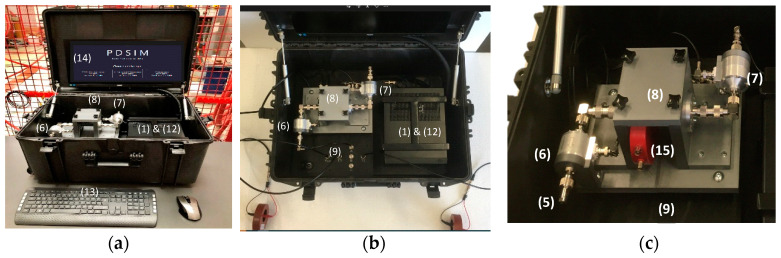
Practical implementation of the calibration setup for qualification of PD analysers: (**a**) front view; (**b**) top view; (**c**) detail view. (1) & (12) AWG & PC, (13) computer keyboard, (14) built-in screen, (5) load resistance for impedance matching, (6) and (7) the two improved HFCT sensors, (8) open test cell, (9) digital recorder, (15) HFCT sensor under characterization.

**Figure 13 sensors-23-05955-f013:**
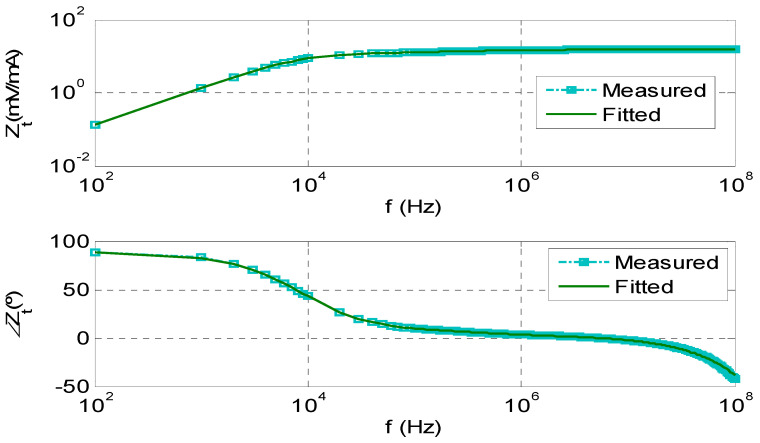
Frequency response of the reference HFCT: amplitude and phase shift of the transfer impedance Z_t_ expressed in (mV/mA) and in degrees (°), and fitting of the transfer impedance using the mathematical expression (12).

**Figure 14 sensors-23-05955-f014:**
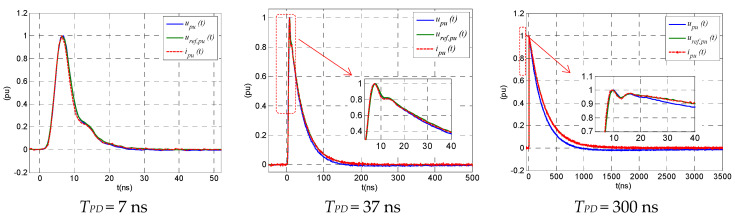
Reconstruction of PD pulses. Blue curve: measured HFCT output, *u_pu_(t)*. Green curve: reference current, *i_ref,pu_(t)*. Red curve: reconstructed signal *i_pu_(t)*.

**Table 1 sensors-23-05955-t001:** Insulation defects causing PD pulses related to HV subsystems.

HV Subsystem in HV Grids	Examples of Insulation Defects Causing PD Pulses
Cable system	Void in XLPE or paper–oil, degraded insulation surfaces of cable joints, false contacts of grounding connections or h.v. conductors.
AIS including porcelain and glass insulators, measuring transformers, disconnectors, switches, surge arresters, etc.	Corona, dirty insulators, cavities in solid insulation, false contacts, floating potential of grounding parts, gas bubbles in measuring transformers, jumping metallic particles in switches.
GIS or GIL	Protrusion, floating potential, jumping particles, metallic particles on insulation surfaces, gas voids in spacers.
HVDC converter	Degraded semiconductor junction, dirty surfaces.
Power transformer	Bubbles in liquid insulations, cavities in paper–oil, moving metallic particles, degraded insulation surfaces, false contacts.

**Table 2 sensors-23-05955-t002:** Characteristic parameters of IDE waveforms to emulate representative PD pulses.

*T_PD_* (ns)	*1/β* (ns)	*1/α* (ns)	*T*_1_*/T*_2_ (ns/ns)	*f_c_* (MHz)	
37.5	11.9	11.9	36.1/53.8	7.5	6.0 ± 1.5
	0.3	35.0	1.7/27.4	4.5	
375	119	119	360/538	0.75	0.59 ± 0.16
	0.3	350	1.8/262	0.43	
75	9.9	44.0	37.8/82.0	3.3	-

**Table 3 sensors-23-05955-t003:** Transfer impedance characteristics of commercial HFCT sensors.

	*HFCT* *Sensor*	*f*_1_ (MHz)	*f*_2_ (MHz)	*Z*_*s*_ mV/mA
** *HVAC* **	**A**	0.12	42	8.6
	**B**	0.07	14	13.4
	**C**	0.06	83	4.8
	**D**	3.01	200	19.3
** *HVDC* **	**E**	0.07	500	15.0

**Table 4 sensors-23-05955-t004:** Reference PD pulse waveforms used for metrological tests.

Reference PD Pulse	*T_PD_* (ns)	1/*α* (ns)	1/*β* (ns)	*β*/*α*	*f_c_* (MHz)
#1	8	4.7	1.06	4.4	30.6
#2	16	9.4	2.13	4.4	15.3
#3	37.5	22.0	4.95	4.4	6.5
#4	75	44.0	9.90	4.4	3.3
#5	150	88.0	19.8	4.4	2.2
#6	375	219.1	50.0	4.4	1.6

**Table 5 sensors-23-05955-t005:** Parameters of damped oscillating PD pulses used for qualification of diagnostic capabilities.

Reference PD Pulse	1/*α* (ns)	1/*β* (ns)	*f* (MHz)	*f_c_* (MHz)	*φ* (°)
#1	44.0	9.90	3	12.2	75
#2	44.0	9.90	6	21.7	60
#3	44.0	9.90	12	43.7	30
#4	44.0	9.90	24	93.9	0

**Table 6 sensors-23-05955-t006:** Capability of the synthetic PD calibrator to generate different charge values depending on the TPD of the pulse.

*T_PD_* (ns)	*q_max_* (pC)	*q_min_* (pC)
8.0	320	2.0
16.0	640	2.0
37.5	1500	2.0
75.0	3000	2.0
150	6000	4.0
375	15,000	10.0

**Table 7 sensors-23-05955-t007:** Charge measurement errors for different PD pulse widths and 200 pC of charge.

α^−1^ (ns)	β^−1^ (ns)	*T_PD_* (5) (ns)	*i_peak_* (mA)	*T_PD_* (Measured)	Error of *T_PD_* %
4.69	1.06	8	25.0	7.9	**−0.75**
9.37	2.13	16	12.5	15.9	**−0.43**
22.0	4.95	37.5	5.33	37.4	**−0.37**
44.0	9.9	75	2.66	74.9	**−0.11**
88.0	19.8	150	1.33	149.1	**−0.58**
219.1	50.0	375	0.533	376.8	**+0.48**

**Table 8 sensors-23-05955-t008:** Linearity error.

Set Charge in Synthetic PD Calibrator (pC)	Charge Measured (pC)	Error (%)	Error (pC)
2500	**2479.3**	**0.8**	**20.7**
1250	**1239.9**	**0.8**	**10.1**
500.0	**496.2**	**0.8**	**3.8**
200.0	**200.0**	**0.0**	**0.0**
100.0	**99.8**	**0.2**	**0.2**
50.0	**49.8**	**0.4**	**0.2**
10.0	**9.7**	**2.7**	**0.3**
2.0	**1.8**	**10.1**	**0.2**
**Linearity = (Max. Error − min Error)/2=**	**50.0 ÷ 2500**	**±0.4%**	
	**2.0 ÷ 200.0**		**±0.2 pC**

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
