# Peer review of "New Synthetic Partial Discharge Calibrator for Qualification of Partial Discharge Analyzers for Insulation Diagnosis of HVDC and HVAC Grids"

_sensors, 2023, doi:10.3390/s23135955_

Round 1

Reviewer 1 Report

1)The abstract and conclusions should not be the same words, they highlight different points, please correct

2)How is the sensitivity of new synthetic PD calibrator for qualification of PD analyzers? Whether to give proof

3)Please calibrate the following grammar form again

1)The abstract and conclusions should not be the same words, they highlight different points, please correct

2)How is the sensitivity of new synthetic PD calibrator for qualification of PD analyzers? Whether to give proof

3)Please calibrate the following grammar form again

Author Response

please check the attachement.

Reviewer 2 Report

The paper presents the study about new synthetic partial discharges calibrator for qualification of PD analyzers for insulation diagnosis of HVDC and HVAC grids. Authors elaborated synthetic PD calibrator in order to the qualification of PD analyzers which use high frequency current transformers. They determined capabilities and measurement errors of the PD analyzers. Final results was that new time parameter "PD Time" was introduced for calibration purpose.

Dear author, thank you very much for interesting paper about PD calibrator for qualification of PD analyzers for insulation diagnosis of HVDC and HVAC grids. I put some comments and questions. PD analysis is still necessary in case of electric power devices, such as power transformers and cable.

Comments:

1. Introduction chapter is well organized. Anyway, I would expect more fundamental information in this chapter about physical meaning of partial discharge pulse.

2. Some negative impacts on electric power devices could be presented, too.

3. Also, authors could consider some information how knowledge about measured PD is able to help in diagnostics of mentioned devices.

4. Please review if all parameters in all formulas are described in main text.

5. What was the object of your investigation? Real high voltage cable. Please add some detail information.

6. General conclusions. the paper is very important and value. it enriches the knowledge about PD diagnostics, especially in frame of calibration, what is very important aspect of mentioned topic. The paper is almost ready to be published, but I expect some small changes which were mentioned before.
